# Secondary through-space interactions facilitated single-molecule white-light emission from clusteroluminogens

Jianyu Zhang [1,6], Parvej Alam[1,6], Siwei Zhang[1,6], Hanchen Shen[1], Lianrui Hu[1], Herman H. Y. Sung [1], Ian D. Williams [1], Jianwei Sun [1], Jacky W. Y. Lam [1✉], Haoke Zhang [2,3,4✉] & Ben Zhong Tang[1,4,5✉]

Clusteroluminogens refer to some non-conjugated molecules that show visible light and unique electronic properties with through-space interactions due to the formation of aggregates. Although mature and systematic theories of molecular photophysics have been developed to study conventional conjugated chromophores, it is still challenging to endow clusteroluminogens with designed photophysical properties by manipulating through-space interactions. Herein, three clusteroluminogens with non-conjugated donor-acceptor structures and different halide substituents are designed and synthesized. These compounds show multiple emissions and even single-molecule white-light emission in the crystalline state. The intensity ratio of these emissions is easily manipulated by changing the halide atom and excitation wavelength. Experimental and theoretical results successfully disclose the electronic nature of these multiple emissions: through-space conjugation for short-wavelength fluorescence, through-space charge transfer based on secondary through-space interactions for long-wavelength fluorescence, and room-temperature phosphorescence. The introduction of secondary through-space interactions to clusteroluminogens not only enriches their varieties of photophysical properties but also inspires the establishment of novel aggregate photophysics for clusteroluminescence.

[1] Department of Chemistry, Hong Kong Branch of Chinese National Engineering Research Center for Tissue Restoration and Reconstruction, and Guangdong-Hong Kong-Macau Joint Laboratory of Optoelectronic and Magnetic Functional Materials, The Hong Kong University of Science and Technology, Clear Water Bay, Kowloon 999077 Hong Kong, China. [2] MOE Key Laboratory of Macromolecular Synthesis and Functionalization, Department of Polymer Science and Engineering, Zhejiang University, Hangzhou 310027, China. [3] ZJU-Hangzhou Global Scientific and Technological Innovation Center, Hangzhou 311215, China. [4] Guangdong Provincial Key Laboratory of Luminescence from Molecular Aggregates, South China University of Technology, Guangzhou 510640, China. [5] School of Science and Engineering, Shenzhen Institute of Aggregate Science and Technology, The Chinese University of Hong Kong, Shenzhen 518172 Guangdong, China. [6] These authors contributed equally: Jianyu Zhang, Parvej Alam, Siwei Zhang. ✉email: chjacky@ust.hk; zhanghaoke@zju.edu.cn; tangbenz@cuhk.edu.cn

Luminescent material not only brightens our life but also leads to far-reaching revolutions in many high-tech areas such as encryption, imaging, and sensor[1–4]. In the past decades, molecular photophysics based on through-bond conjugation (TBC) has been established to guide the design of efficient and multi-functional organic luminophores, and structures with extended π-conjugation are recognized as the prerequisite for efficient luminescence performance[5–7]. However, some non-conjugated molecules have attracted significant attention recently due to their ability to emit visible light in the aggregate state[8–10]. For example, non-conjugated and non-aromatic poly(amidoamine) dendrimers and succinimide derivatives show strong blue and green emission in the clustering state, respectively[11–13]. This non-conventional luminescence is termed clusteroluminescence (CL), and the luminophores with such property are known as clusteroluminogens (CLgens)[14–16]. Compared with traditional luminophores with conjugated aromatic rings, CLgens possess better flexibility and processability. Besides, they show good biocompatibility compared with inorganic materials due to the low toxicity and better degradability, thus playing as promising luminescent materials for biological applications[17,18].

However, different from traditional conjugated luminophores, the TBC-based theories usually fail to explain the photophysical behaviors of CLgens. Thus, there is an urgent need to build new photophysical theories for CL[19]. Previous reports indicate that the through-space interactions (TSI) and electron coupling/delocalization between spatially separated units play an essential role in the visible emission of non-conjugated CLgens[20–22]. For instance, 1,1,2,2-tetraphenylethane, in which its phenyl rings are isolated from each other by saturated carbons, shows sky blue emission in the solid state due to the excited-state electron overlap of the phenyl rings[23]. Tang et al. also demonstrated the possibility of regulating TSI by introducing electron-donating and withdrawing groups to not only adjust the electron density but also affect the rigidity of excited-state geometry[24,25]. Although several achievements have been received in TSI-related studies, it is still a big challenge to improve the photophysical performance of CLgens by manipulating the TSI at the molecular level.

In contrast, many effective strategies have been developed to modify the electronic structures of conjugated chromophores according to theories of TBC, such as excited-state intramolecular proton transfer[26], twisted intramolecular charge transfer[27–29], and photoinduced electron transfer[30]. Because of the diversified electronic structures and properties, it is easy to fabricate single-molecule white-light emission (SMWLE) in conjugated molecules via multiple emissions[31–35]. As shown in Fig. 1a, it is a typical strategy to generate multiple emissions from conjugated chromophores with moderate donor-acceptor structures through the incorporation of locally excited (LE) state, charge transfer (CT) state, and room-temperature phosphorescence (RTP)[36–39]. However, the rigid structures of such molecules make it difficult to regulate the intensity ratio of these multiple emissions. On the other hand, the non-conjugated and flexible structures of CLgens endow them with sensitive response to internal and external stimuli, which is beneficial to the generation of multiple emissions and SMWLE. Unfortunately, it is still a thorny issue to realize SMWLE in CLgens by manipulating their TSI due to the limited recognition on TSI-related processes.

In this work, we developed a feasible strategy to achieve multiple emissions and SMWLE from CLgens with totally isolated phenyl rings (Fig. 1b, c). Three compounds, namely TPMI-X (X = Cl, Br and I), were synthesized based on triphenylmethanamine (TPMA) and halogen-substituted phenylmethanimine (PMI-X, X = Cl, Br and I), and fully characterized (Supplementary Figs. 3–6 and 11–14). Photophysical measurements revealed two fluorescent emissions and one RTP in the crystalline state.

Meanwhile, the intensity ratio of these multiple peaks can be manipulated by both the internal heavy-atom effect and external excitation wavelength. Additionally, theoretical calculations indicated the important role of through-space conjugation (TSC) of TPMA and secondary TSI which performs the through-space charge transfer (TSCT) from TPMA to PMI-X in producing SMWLE. It is noteworthy that this work not only provides a general strategy to achieve SMWLE from unconventional CLgens but also enriches the photophysical behaviors and mechanistic understanding of CL and TSI.

## Results

**Photophysical properties**. The absorption spectra of the newly synthesized compounds were first measured (Fig. 1d). All the compounds exhibited absorption maximums with a wavelength of shorter than 300 nm in both THF solution and solid state, which were assigned to the absorption of the PMI-X group and isolated phenyl rings. This result suggested the non-conjugated nature of these compounds. On the other hand, no photoluminescence (PL) signals were detected in their pure THF solutions and even THF/$H_2O$ (1:9, $v/v$) mixtures where aggregates were formed (Supplementary Figs. 20 and 21). This was presumably due to the flexible structure of TPMI-X, where their vigorous molecular motions could not be effectively restricted even in the aggregate state. Bulk crystals of the compounds were easily obtained through recrystallization from their water/ethanol mixtures and were highly emissive under 365 nm UV irradiation, suggesting their characteristics of aggregation-induced emission (AIE)[40–42]. Figure 2a–c and Supplementary Fig. 22 showed their PL properties in the crystalline state. All the spectra showed two distinct emission peaks in the visible range, where their relative intensity was excitation dependent. For TPMI-Cl, while the peak at around 400 nm was dominant, another low-intensity shoulder peak was located at 490 nm. The nature of these two emissions was assigned to fluorescence as suggested by their nanosecond lifetimes (Fig. 2d). The lifetime at 490 nm was longer than that at 400 nm, indicating the different emitting species of these two peaks. The TPMI-Cl crystal exhibited bluish-white emission with an absolute quantum yield (QY) of 34.8% at an excitation wavelength ($\lambda_{ex}$) of 370 nm. When the halogen atom was replaced by different elements from Cl to Br and I, the long-wavelength peak gradually intensified and became dominant, and its position also red-shifted to 500 nm (TPMI-Br) and 506 nm (TPMI-I), respectively (Fig. 2b, c). Interestingly, TPMI-Br showed a perfect emission balance and demonstrated white-light emission with a QY of 28.3% at $\lambda_{ex}$ of 370 nm. The long-wavelength emission was dominant in TPMI-I, and it overall showed greenish-white emission with a QY of 14.6%. It is noteworthy that the intensity ratio ($I_L/I_S$) of the long- ($I_L$) and short-wavelength emission ($I_S$) of each compound decreased progressively along with the increased $\lambda_{ex}$ (Fig. 2g). This could be explained by their excitation spectra, where the long-wavelength emission showed a relatively shorter excitation peak than the short-wavelength one, so that the long-wavelength emission was more easily excited by the shorter $\lambda_{ex}$ than the longer ones (Fig. 2h and Supplementary Fig. 23). Besides, under the same $\lambda_{ex}$, the ratio of $I_L/I_S$ decreased in the order from TPMI-I to TPMI-Br and then to TPMI-Cl. After grinding the bulk crystals of TPMI-Br, the amorphous powder was formed, and its PL spectra were recorded (Fig. 2i and Supplementary Fig. 24). Its spectra displayed a broad peak where the short- and long-wavelength emissions were mixed. The associated QYs were much lower than those in the crystalline state and changed slightly at different $\lambda_{ex}$. These results indicated that the PL performance was closely related to the intra-/intermolecular interactions in the crystalline state and the halide substituents.

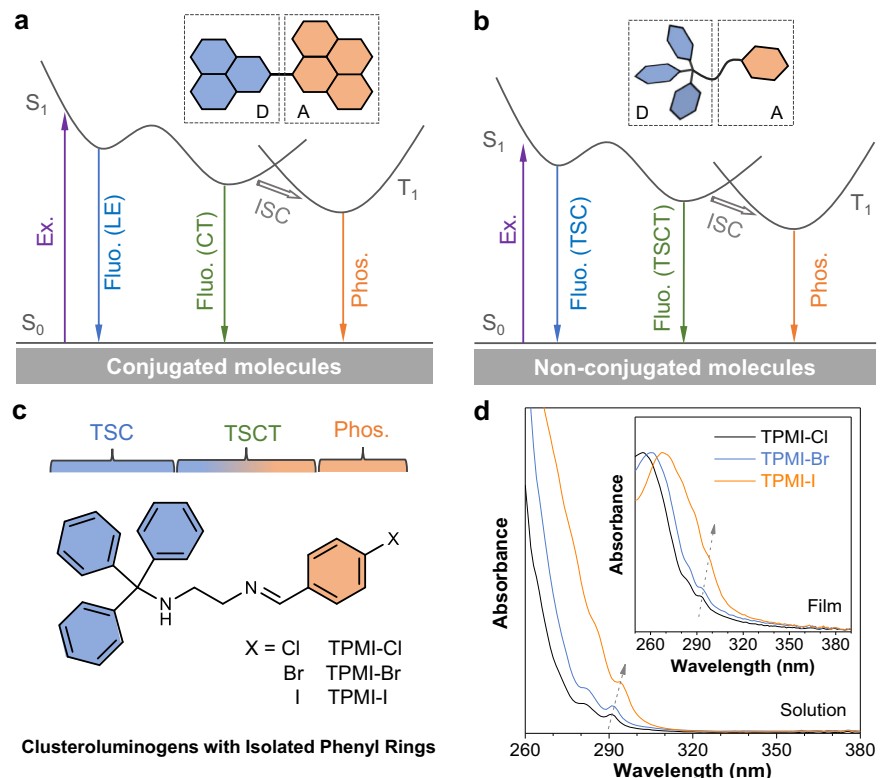

**Fig. 1 Design strategy of non-conjugated molecules with multiple emissions. a** A typical mechanism of multiple emissions from organic molecules with extended conjugated structures. **b** Proposed multiple emissions from non-conjugated molecules with isolated phenyl rings. **c** Designed molecules based on triphenylmethanamine and phenylmethanimine groups connected by a flexible ethyl chain. **d** Absorption spectra of TPMI-Cl, TPMI-Br, and TPMI-I in THF solution and solid film (inset), respectively. Ex. excitation, Fluo. fluorescence, Phos. phosphorescence, LE locally excited, CT charge transfer, TSC through-space conjugation, TSCT through-space charge transfer, ISC intersystem crossing, D electron donor, A electron acceptor.

Additionally, RTP was also observed from these three compounds, as depicted in Fig. 3. All compounds exhibited similar RTP peaks at around 560–600 nm, although their fluorescent properties were different. The lifetime of RTP was 24 ms in TPMI-Cl, which was shortened to 17 ms in TPMI-Br and 5.1 ms in TPMI-I, according to the law of heavy-atom effect[39,43,44]. At the cryogenic temperature of 77 K, the phosphorescent emission became stronger, shifted to the bluer region, and showed a longer lifetime (Fig. 3d and Supplementary Fig. 25). Thus, the above results verified the ability of the present compounds as CLgens with multiple emissions and even SMWLE, despite their non-conjugated structures.

**Intramolecular through-space interactions**. To disclose the mechanism and origin of the unusual multiple emissions from the present CLgens, two model compounds, namely, $N$-methyl-triphenylmethanamine (Me-TPMA) and halogen-substituted $N$-methyl-phenylmethanimine (Me-PMI-X, X = Cl, Br and I), were synthesized and characterized (Supplementary Figs. 7–10 and 15–18). As expected, the UV spectrum of Me-TPMA showed a peak maximum at 264 nm, which corresponded to the absorption of isolated phenyl rings (Supplementary Fig. 26). Interestingly, Me-TPMA showed bright visible emission with maximal wavelength ($\lambda_{em}$) at 445 nm and a QY of 7.2% (Fig. 4a). The corresponding excitation maximum was located at 366 nm, which was ascribed to the TSC of the isolated phenyl rings, and such interaction could be directly visualized from the hole-electron analysis with spatial electron communication (Fig. 4b)[24]. Its emission wavelength was very close to the short-wavelength emission of these three CLgens. On the other hand, Me-PMI-X

showed excitation and emission maximums at around 280 and 300 nm, respectively, and the hole-electron analysis suggested a $(\pi,\pi^*)$ transition (Fig. 4c, d and Supplementary Fig. 27). Meanwhile, the emission wavelength of Me-PMI-X was much shorter than that of TPMI-X. Based on these considerations, it is speculated that the short-wavelength emission of TPMI-X should originate from the TSC of the TPMA moiety.

In consideration of the non-conjugated structure of TPMI-X, the possibility of intermolecular CT in the solid state was firstly evaluated by evenly mixing Me-TPMA and Me-PMI-X with a molar ratio of 1:1. Taking the solid mixture of Me-TPMA/Me-PMI-Br for example, its PL spectra showed excitation-independent and broad peaks with $\lambda_{em}$ around 533 nm (Fig. 4e), which is longer than the observed long-wavelength emission from pure TPMI-Br crystals of 500 nm. The excitation spectra with a maximal peak at 463 nm also indicated the feature of intermolecular CT (Fig. 4f), which was also much longer than that of TPMI-Br with an excitation maximum at 364 nm (Fig. 2h). Besides, another two mixtures of Me-TPMA/Me-PMI-Cl and Me-TPMA/Me-PMI-I showed similar emission and excitation spectra. (Supplementary Fig. 28). As a result, the long-wavelength emission and excitation maximum of TPMI-X were blue-shifted than these mixtures. Therefore, it was believed that the long-wavelength emission from TPMI-X was mainly contributed by the intramolecular behaviors of two separated units, while the intermolecular CT process may play a minor role. In addition, the single-crystal X-ray diffraction technique was utilized to investigate their crystal structures and intermolecular interactions (Supplementary Table 1 and Supplementary Figs. 29–31). All the molecules exhibited similar head-to-tail arrangements where each halogen atom formed two halogen···H bonds with the TPMA unit

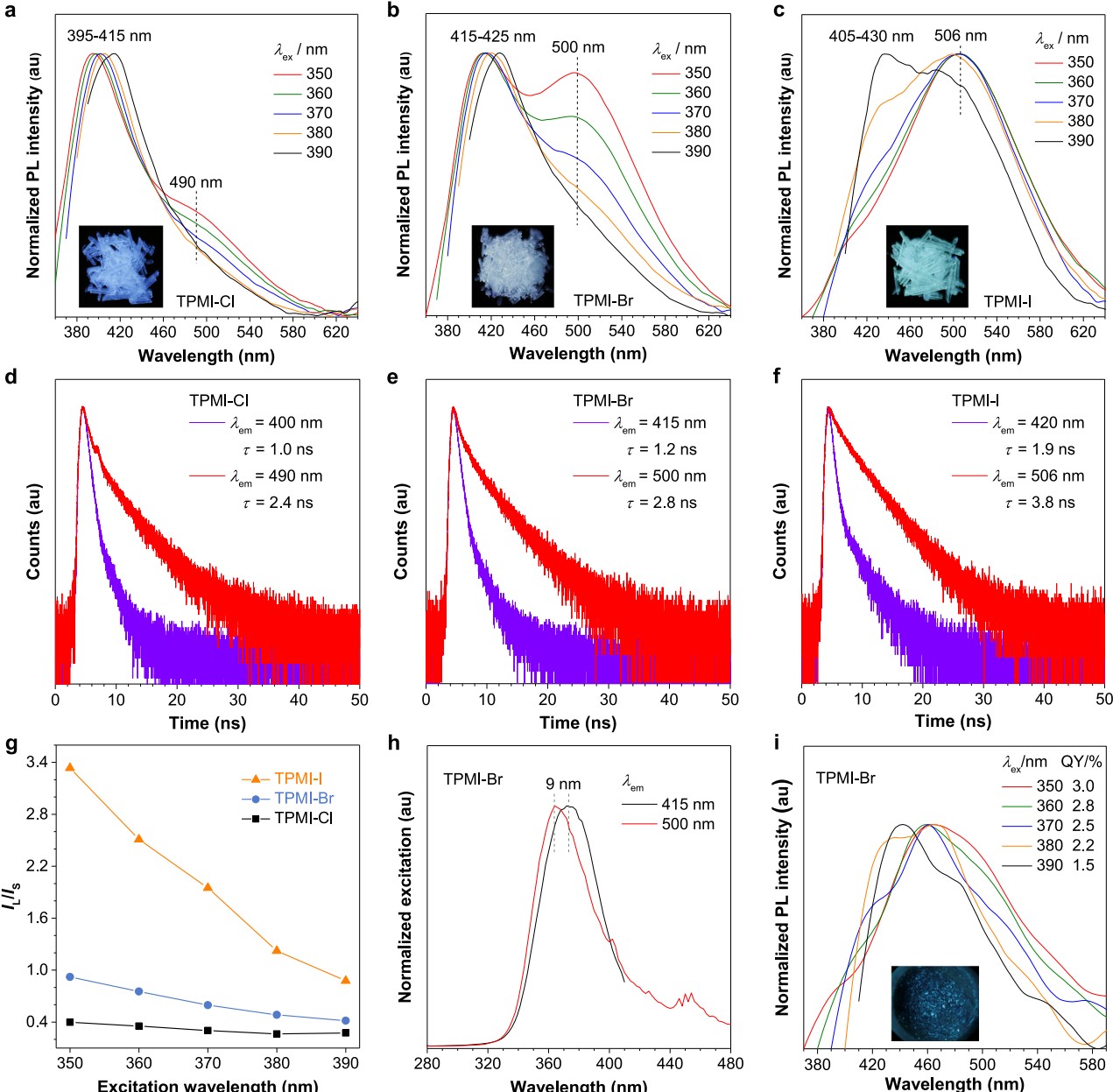

**Fig. 2 Photophysical properties of the designed compounds.** Normalized photoluminescence (PL) spectra of (**a**) TPMI-Cl, (**b**) TPMI-Br, and (**c**) TPMI-I in the crystalline state at different excitation wavelengths ($\lambda_{ex}$); Insets: photos of the crystals taken under irradiation of a 365 nm UV lamp. Time-resolved photoluminescence decay curves and lifetimes ($\tau$) of (**d**) TPMI-Cl, (**e**) TPMI-Br, and (**f**) TPMI-I taken at their corresponding emission maximums ($\lambda_{em}$). **g** Plots of the relative PL intensity ($I_L/I_S$) versus the $\lambda_{ex}$, where $I_L$ and $I_S$ are the intensity maximum of the long-wavelength emission and short-wavelength emission, respectively. **h** Excitation spectra of TPMI-Br in the crystalline state taken at $\lambda_{em}$ of 415 nm and 500 nm. **i** Normalized PL spectra and quantum yields (QY) of TPMI-Br in the amorphous state at different $\lambda_{ex}$. Insets: photo of the TPMI-Br in the amorphous state taken under irradiation of a 365 nm UV lamp.

of another molecule, and no strong intermolecular interactions were observed. Hirshfeld surface analysis based on the crystal structures was performed to quantitatively evaluate their intermolecular interactions (Fig. 4g)[45]. These weak intermolecular C···H, H···H, and halogen···H interactions accounted for more than 95% to total intermolecular interactions, which mainly contributed to the mechanism of restriction of intramolecular motion (RIM). Therefore, the above experimental and theoretical results excluded the possibility that multiple emissions of TPMI-X originated from intermolecular photophysical processes, but verified that the intramolecular interactions should play major roles in their photophysical behaviors.

To further unveil the electronic nature of these emission peaks in TPMI-X, TPMI-Br was selected as an example to be studied with the help of theoretical calculation. From the hole-electron analysis of TPMI-Br in the excited state, it was apparent that the hole and electron were mainly distributed on the TPMA moiety, and the hole-electron distribution was almost the same as that of Me-TPMA showing TSC (left panel in Figs. 5a and 4b). Besides, the calculated energy gap ($E_{em}$ (cal.)) of 3.43 eV was comparable to that of the short-wavelength emission at 410 nm ($E_{em}$ (exp.) = 3.00 eV). The combined experimental and theoretical results supported the TSC as the origin of the short-wavelength emission from TPMI-Br.

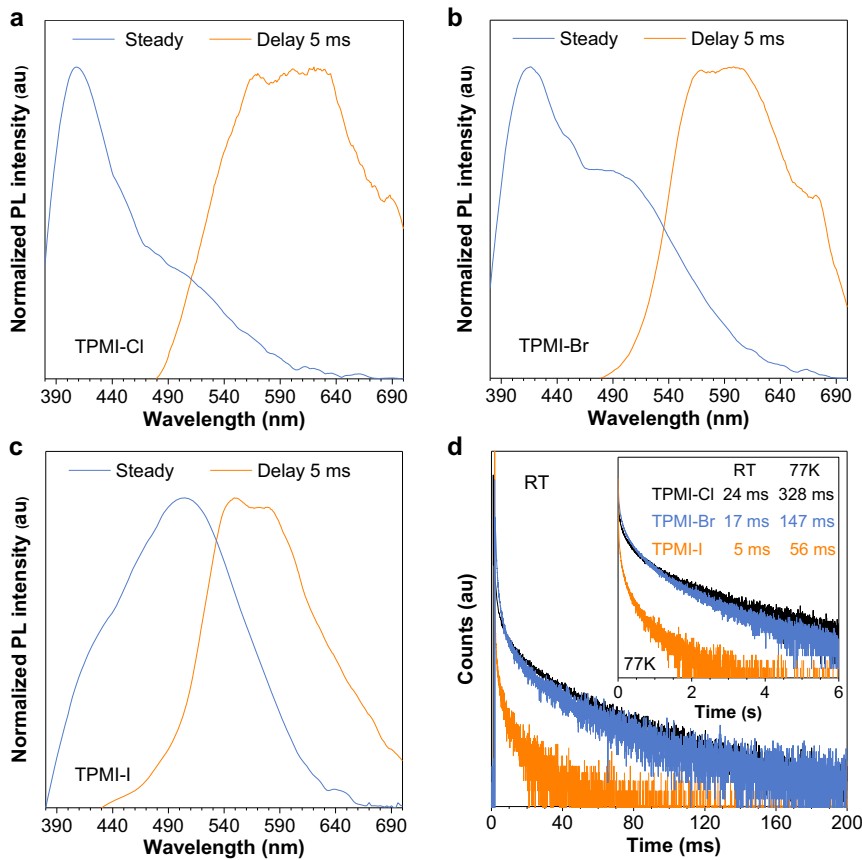

**Fig. 3 Room-temperature phosphorescence.** Normalized steady and delayed photoluminescence (PL) spectra of (**a**) TPMI-Cl, (**b**) TPMI-Br, and (**c**) TPMI-I in the crystalline state. **d** Time-resolved PL decay curves of the compounds at emission maximum of 570 nm at room temperature and 560 nm at 77 K (inset), respectively. Excitation wavelength is 375 nm.

Then, the intramolecular CT process between the separated TPMA and PMI-Br moieties was further investigated. As shown in the middle panel of Fig. 5a, another low-lying excited state with a CT feature was mapped where the hole and electron were distributed on the TPMA and PMI-Br moieties, respectively. Meanwhile, the calculated energy gap of 2.44 eV was almost the same as that of the long-wavelength emission at 500 nm ($E_{em}$ (exp.) = 2.48 eV), suggesting the accurate simulation of geometry and electronic transition behavior. Since the donor (TPMA) and acceptor (PMI-Br) units were separated by a flexible and saturated ethyl linker, this transition was regarded as another important through-space behavior known as through-space charge transfer (TSCT). The effect of TSCT is widely studied and proved as another mechanism for luminescence from conjugated organic molecules, especially in the field of thermally activated delayed fluorescence and RTP[46–51]. It is noteworthy that, different from the reported TSCT process caused by the spatially electronic interactions between the TBC-based donor and acceptor units, the TSCT in TPMI-Br is based on the non-conjugated donor with TSC. This particular interaction was termed secondary TSI. The TSCT was generated from a TSC-based CLgens, which was responsible for the long-wavelength fluorescence from TPMI-Br, TPMI-Cl, and TPMI-I. Besides, the amino group of the TPMA unit played as a "bridge" to facilitate the secondary TSI as well as increase the electron-donating ability of the TPMA unit.

It is noteworthy that TSCT state with the (n,π*) feature may quench the emission of some fluorogens in the solution state due to spatially weak electron overlap and low oscillator strength[52,53]. Since the TSCT effect was not reflected in the absorption spectra

of the three compounds (Fig. 1d), the excited-state intramolecular motion should play another crucial role in enhancing electron communication and facilitating this TSCT transition[54,55]. Accordingly, the distance between two nitrogen atoms ($N_1$-$N_2$) and the dihedral angle of the two separated parts ($\angle N_1$-$C_1$-$C_2$-$N_2$) were tracked along with the optimization process from the optimized ground state to the excited TSCT state (Fig. 5b, c). Results showed that the $N_1 - N_2$ distance became shorter from the ground-state geometry of 2.93 Å to the excited-state geometry of 2.80 Å, and the dihedral angle also decreased from 65° to 50°. These variations indicated that the electron overlapping of the two separated groups was enhanced in the excited state, which formed an efficient (n,π*) transition channel for TSCT. TPMI-Cl and TPMI-I displayed similar hole-electron distribution and geometry change (Supplementary Figs. 33 and 34 and Supplementary Table 2), suggesting that the secondary TSI contributed a lot to realizing the long-wavelength fluorescent emission.

Besides, the origin of RTP was also investigated. Both the triplet spin density and hole-electron distribution of TPMI-Br indicated that its phosphorescence was aroused by the (n,π*) transition of PMI-Br moiety (right panel in Fig. 5a, and Supplementary Fig. 35). Meanwhile, the calculated triplet-state energy gap ($E_{em}$ (cal.)) of 2.36 eV was close to the observed phosphorescence at 570 nm ($E_{em}$ (exp.) = 2.18 eV). However, the triplet-state (n,π*) transition of TPMI-Br was different from that of Me-PMI-Br with a triplet (π,π*) transition (Supplementary Fig. 36). This might be due to the ethyl chain of TPMI-Br that ruined its planarity and altered the energy level of n and π orbitals. Besides, the strength of intersystem crossing (ISC) of TPMI-Br was also evaluated (Fig. 5d and Supplementary Fig. 37).

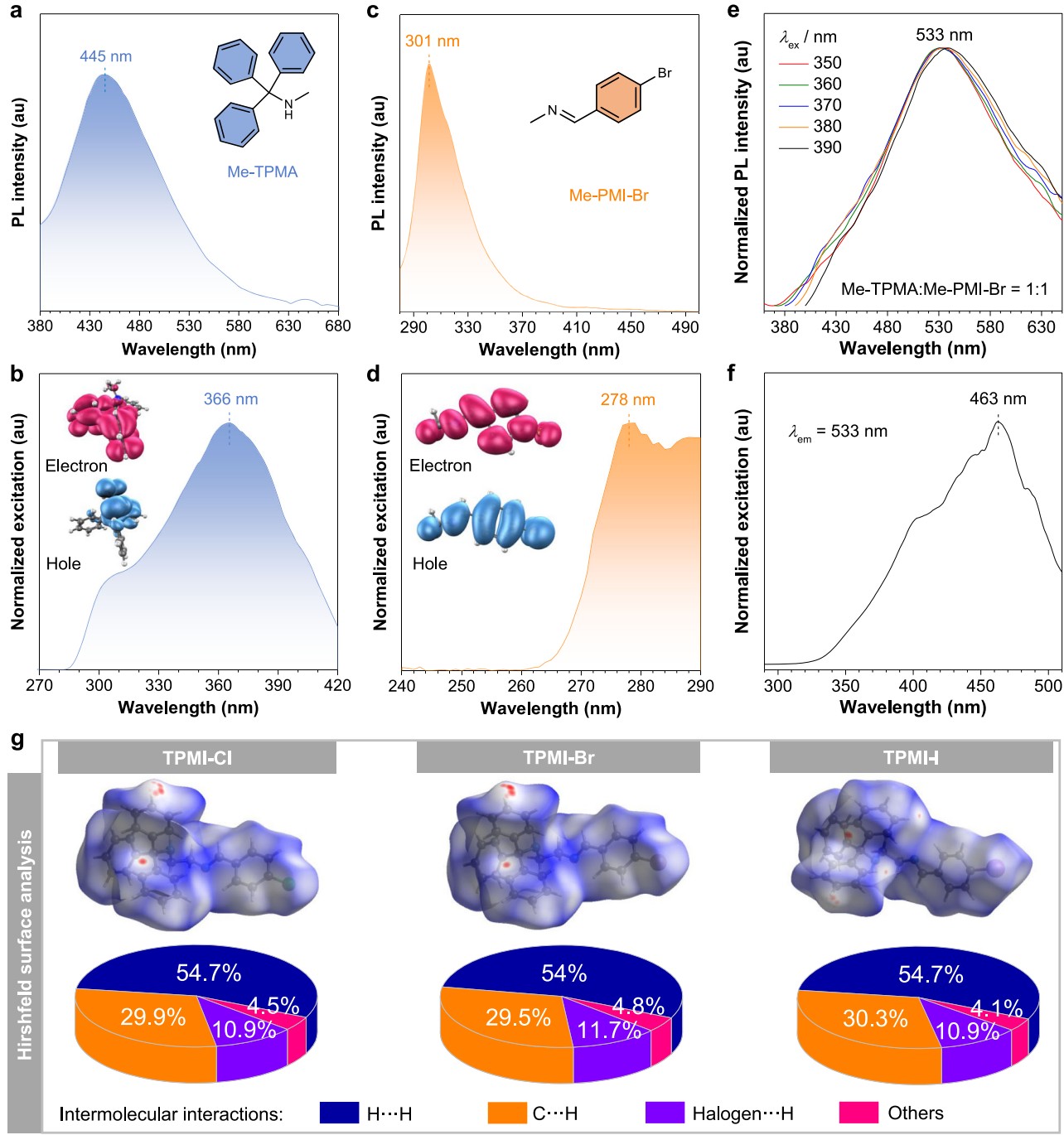

**Fig. 4 Photophysical properties of the model compounds and intermolecular interaction analysis. a** Photoluminescence (PL) spectra of Me-TPMA in the solid state under an excitation wavelength ($\lambda_{ex}$) of 365 nm. **b** Excitation spectra of Me-TPMA with an emission wavelength ($\lambda_{em}$) of 445 nm. Inset: hole-electron analysis of Me-TPMA. **c** PL spectra of Me-PMI-Br under a $\lambda_{ex}$ of 280 nm. **d** Excitation spectra of Me-PMI-Br with a $\lambda_{em}$ of 301 nm. Inset: hole-electron analysis of Me-PMI-Br. **e** Normalized PL spectra of the Me-TPMA/Me-PMI-Br mixture with a molar ratio of 1:1 in the solid state under different $\lambda_{ex}$. **f** Excitation spectra of the Me-TPMA/Me-PMI-Br mixture in the solid state taken at a $\lambda_{em}$ of 533 nm. **g** Hirshfeld surface analysis (mapped over $d_{norm}$), and proportions of intermolecular C···H, H···H, halogen···H, and other interactions to the total intermolecular interactions based on their crystal structures.

It showed a small energy gap of 0.067 eV between the singlet and triplet state and comparatively large constants ($\xi$) of spin-orbital coupling, resulting in efficient ISC and RTP in the crystalline state.

To better understand the different photophysical behaviors of TPMI-Br in the solution and crystalline states, its molecular motions in the two states were investigated (Fig. 5e). Firstly, TPMI-Br was optimized as a single molecule under a polarizable

continuum model with tetrahydrofuran as solvent. As expected, it showed strong intramolecular motions in the solution phase, as evidenced by the large value (1.021 Å) of root-mean-square deviation (RMSD) which described the change of atomic positions between the geometries of the optimized ground state (blue color) and excited TSCT state (red color). The poorly overlapped geometries also demonstrated its flexible structure and vigorous intramolecular motions, resulting in totally

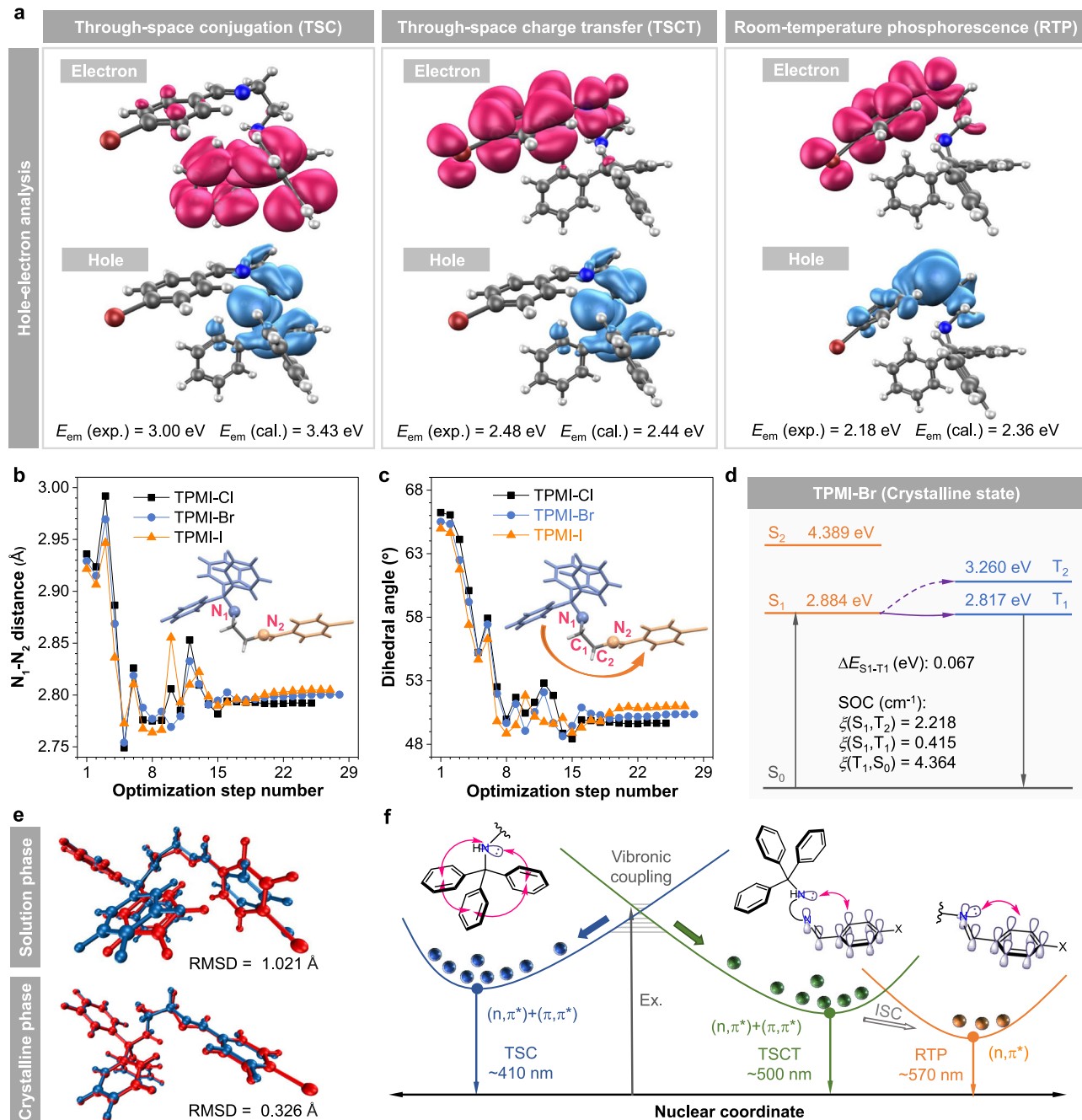

**Fig. 5 Proposed mechanism of multiple emissions of designed clusteroluminogens. a** Hole-electron analysis of TPMI-Br based on the corresponding optimized geometries calculated at B3LYP-D3/Def2-SVP level. $E_{em}$ (exp.) experimental energy gap, $E_{em}$ (cal.) calculated energy gap. Plots of (**b**) $N_1-N_2$ distance and (**c**) dihedral angle of $\angle N_1-C_1-C_2-N_2$ versus optimization steps from the ground state to the excited TSCT state. Inset: structure of the designed clusteroluminogens with highlighted atoms. **d** Energy gap between singlet and triplet states ($\Delta E_{S1-T1}$) and spin-orbital coupling (SOC) constants ($\xi$) of TPMI-Br. **e** Overlaps of the optimized ground-state (blue color) and singlet TSCT (red color) geometries of TPMI-Br in the solution and crystalline phases, respectively. The root-mean-square deviation (RMSD) of atomic positions was calculated to evaluate the strength of intramolecular motions in two different phases. **f** Potential energy surface and electronic behaviors of the designed clusteroluminogens to realize multiple emissions. Ex. excitation, TSC through-space conjugation, TSCT through-space charge transfer, RTP room-temperature phosphorescence, ISC intersystem crossing.

quenched emission even in the aggregate state. Besides, the crystalline state was simulated based on the crystal structure using the ONIOM model with a combined quantum mechanics and molecular mechanics (QM/MM) approach. The central molecule was optimized as the high layer, while the surrounding molecules were frozen as the low layer (Supplementary Fig. 38). The results showed that the intramolecular motions were restricted to a confined level in the crystalline state with a small RMSD value of

0.326 Å, which suggested the RIM as the mechanism for the AIE effect and crystalline-state emission of TPMI-Br[56–58]. The confined RIM mechanism simultaneously inhibited the non-radiative decay and promoted the formation of secondary TSI and long-wavelength emission from the TSCT state.

According to the above results, a comprehensive picture of multiple emissions from the CLgens was drawn and illustrated (Fig. 5f). After photoexcitation from the ground state, two

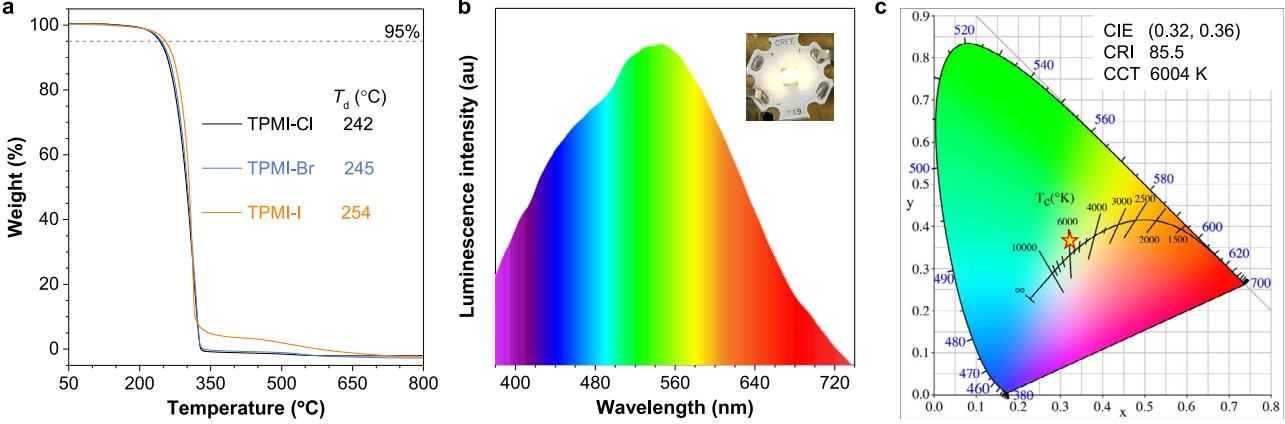

**Fig. 6 Thermal stability and fabrication of a white-light emitter. a** Thermogravimetric analysis and decomposition temperature ($T_d$) of the three targeted compounds recorded under N$_2$ gas at a heating rate of 10 °C/min. **b** The luminescence spectra and digital photo of the fabricated white-light emitter by coating microcrystals of TPMI-Br and epoxy resin glue on a commercial UV chip. **c** The Commission Internationale de L'Eclairage (CIE) chromaticity coordinate of the fabricated emitter. CRI color rendering index, CCT correlated color temperature.

relaxation pathways, namely TSC and TSCT states, were feasible due to their overlapped excitation spectra and strong vibronic coupling (Fig. 2h). The TSC state was related to the spatial electron communication of the three isolated phenyl rings in TPMA moiety and showed blue emission at around 410 nm. On the other hand, the TSCT state relied on the (n,π*) transition of the two separated parts, which formed secondary TSI via the excited-state intramolecular motion and gave the long-wavelength emission at 500 nm with green color. Due to halogen atoms and enhanced intersystem crossing, the RTP from the PMI-X moiety of TPMI-Cl, TPMI-Br, and TPMI-I at around 570 nm was also detected. These three components composed multiple emissions and SMWLE of the designed CLgens.

**White-light emitters.** In light of the particular multiple emissions of the three CLgens and their excellent thermal stability (Fig. 6a), a prototype of an organic white-light emitter was fabricated by coating a mixture of TPMI-Br microcrystals and epoxy resin glue on a commercial UV chip (Fig. 6b, c). These microcrystals didn't dissolve during fabrication and were embedded into the glue (Supplementary Fig. 40). Within the current range considered, the fabricated emitter displayed bright white-light emission with a Commission Internationale de L'Eclairage (CIE) chromaticity coordinate of (0.32, 0.36), which was close to the value of standard white light. The luminescence spectra fitted well with the PL spectra of the TPMI-Br crystal. Besides, the color of the fabricated emitter exhibited high quality with a color rendering index of 85.5 and correlated color temperature of 6004 K, suggesting its potential for daylight illumination and display. Similarly, the fabricated emitters based on TPMI-Cl and TPMI-I showed similar luminescence with cold-white and daylight colors, respectively (Supplementary Figs. 41 and 42). It is worth noting that the poor ability of charge transport due to the non-conjugated structures hinders their application as an emissive layer in organic light-emitting diodes, although they displayed multiple emissions and good color index.

## Discussion

In this work, three non-conjugated CLgens, namely TPMI-X (X = Cl, Br and I), based on flexible ethyl-linked TPMA (donor) and PMI-X (acceptor) were synthesized, and their photophysical properties were systematically investigated. All the compounds displayed multiple emissions consisting of two fluorescence components and one RTP unit in the crystalline state and the

highest absolute QY of 34.8%. The intensity ratio of these multiple emissions was easily manipulated by changing the halide substituent and external excitation wavelength, demonstrating their tremendous potential in producing SMWLE. Theoretical calculation and crystal-structure analysis proved that the short-wavelength fluorescence stemmed from the TSC of TPMA moiety. Because of the non-conjugated donor-acceptor structure, the peculiar intramolecular TSCT interaction as the secondary TSI was formed between the TSC-based TPMA and PMI-X to promote the long-wavelength fluorescence. Halogen-atom assisted RTP was observed as the third emission peak. In addition, the multiple intermolecular interactions provided a rigid environment to restrict the intramolecular motions of the flexible structures and stabilize the excited-state geometries, leading to the highly efficient multiple emissions. A prototype of a white-light emitter based on TPMI-Br was fabricated with a CIE coordinate of (0.32, 0.36) and CCT value of 6004 K, indicating its excellent potential for daylight illumination and display.

We innovatively introduced the concept of secondary through-space interactions of CLgens. Similar to the important role of secondary structure in protein bioactivity, the through-space charge transfer based on non-conjugated moiety with TSC endows CLgens with versatile and controllable photophysical behaviors, which will be a promising and significant theory of aggregate photophysics for non-conjugated CLgens. It is anticipated that, because of secondary TSI, clusteroluminescence will become an influential luminescent form not only in fundamental research but also in the cutting edge of technology development and application.

## Methods

**Materials**. All chemicals and reagents were purchased from commercial sources. Triphenylmethyl chloride (99%), 4-chlorobenzaldehyde (98%) and 4-chlorobenzaldehyde (97%) were purchased from J&K Scientific Ltd. Ethylenedia-mine (≥99%), and aminomethane (33 wt. % in absolute ethanol) were obtained from Merck KGaA (Sigma-Aldrich), Darmstadt, Germany. 4-Bromobenzaldehyde (>97%) was purchased from Tokyo Chemical Industry Co. Ltd. All the final products used in experiments were purified by silica gel column or recrystallization at least three times. The purification of all designed samples was carefully checked by high-performance liquid chromatography. Tetrahydrofuran (THF) used in experiments was distilled from sodium benzophenone ketyl under nitrogen gas.

**Instrumentation**. High-performance liquid chromatography (HPLC) was measured on a Waters 2487 (600E) equipment with a column of Poroshell 120 (EC-C18, 2.7 μm, 4.6 × 150 mm), using acetonitrile/water mixture (ratio = 9:1, $v/v$) with a flow rate of 1.0 mL/min. Nuclear magnetic resonance (NMR) spectra were carried out on a Bruker AVIII 400 MHz NMR spectrometer equipped with a Dual Probe,

using deuterated chloroform ($CDCl_3$) as the solvent. High-resolution mass spectra (HRMS) were measured on a GCT premier CAB048 mass spectrometer manipulated in a GC-TOF module with chemical ionization (CI). Ultraviolet-visible (UV-Vis) spectra were collected on a Varian Cary 50 Conc UV-Visible Spectrophotometer with Peltier. Photoluminescence (PL) spectra and their lifetimes at room temperature and 77 K were recorded on an Edinburgh FLS980 Spectrometer. Absolute quantum yield (QY) was collected on a Hamamatsu Quantum Yield Spectrometer C11347 Quantaurus. Single-crystal X-ray diffraction (XRD) was carried out on a Rigaku Oxford Diffraction (SuperNova) with Atlas diffractometer (Cu Kα (λ = 1.54184 Å)). The single-crystal structures were solved with Olex2 software. Dynamic light scattering (DLS) measurements for the diameter of aggregates were measured on a Malvern Zetasizer Nano ZS equipment at room temperature. All digital photos of crystals and amorphous compounds were recorded on a Canon EOS 60D camera.

**Computational details**. All molecules were optimized using the density functional theory (DFT) method with B3LYP density functional and Def2-SVP basis set. Grimme's DFT-D3 correction was utilized to better describe London-dispersion effects and long-range inter/intramolecular charge transfer processes. Besides, analytical frequency calculations were also carried out at the same level of theory to confirm the local minimum point of the optimized structures. Conformational searches were also performed using the GMMX method with the MMFF94 force field to identify the minimum energy conformation. Time-dependent density functional theory (TD-DFT) was performed at the same level of theory to calculate optimized singlet-state geometries and energy levels. Meanwhile, the triplet state was optimized using the unrestricted density functional theory (UKS-DFT) method. The polarizable continuum model (PCM) and self-consistent reaction field (SCRF) with tetrahydrofuran as solvent were used to calculate single-molecule geometries in the solution phase. Geometries and energy levels of the crystalline phase were performed based on the combined quantum mechanics and molecular mechanics (QM/MM) model selected from the crystal packing structures. The central molecule was treated as the QM part and was optimized at (TD) B3LYP-D3/Def2-SVP level, while the surrounding molecules were frozen to act as the MM part with the universal force field (UFF). All the above quantum chemical calculations were carried out using Gaussian 09 program (Revision D.01). Besides, to evaluate the intersystem crossing efficiency of the three compounds, the spin-orbital coupling (SOC) constants were calculated based on optimized singlet-state geometries using the BDF package. The Hirshfeld surfaces and decomposed fingerprint plots were mapped using CrystalExplorer 17.5 package, and the hole-electron analysis was displayed using the IQmol molecular viewer package.

**Root-mean-square deviation**. Root-mean-square deviation (RMSD) was calculated to evaluate the strength of intramolecular motions between the optimized ground-state and excited-state geometries. The value of RMSD is calculated by Eq. (1):

$$\text{RMSD} = \sqrt{\frac{1}{N}\sum_i^{natom}[(x_i - x_i')^2 + (y_i - y_i')^2 + (z_i - z_i')^2]} \qquad (1)$$

Where $i$ is the number of all atoms, and $x$, $y$, and $z$ are the cartesian coordinates of the optimized ground-state geometry ($x_i$, $y_i$, $z_i$) and excited-state geometry ($x_i'$, $y_i'$, $z_i'$), respectively.

**White-light emitters**. The microcrystals of organic compounds (TPMI-X, X = Cl, Br, and I) were mixed with epoxy resin glue at 5 wt.%, and the mixture was manually stirred with a stainless-steel bar for ~5 min until it was uniform. Then the mixture was coated on the surface of a commercial UV chip until the reflective cavity was filled. These chips were purchased from the Shenzhen Looking Long Technology Co., Ltd. (Shenzhen, China) with a power of 1 W and an emission maximum of 310 nm. The performance of the fabricated emitters was recorded using an auto-temperature LED optoelectronic analyzer with a temperature controller (ATA-500, EVERFINE, China).

## Data availability

All the data supporting the findings in this work are available within the manuscript and Supplementary Information file, and available from the corresponding authors upon request. The X-ray crystallographic coordinates for structures reported in this work have been deposited at the Cambridge Crystallographic Data Centre (CCDC) under deposition numbers of 2122661 (TPMI-Cl), 2122660 (TPMI-Br), and 2122662 (TPMI-I). These data can be obtained free of charge from The Cambridge Crystallographic Data Centre via www.ccdc.cam.ac.uk/data_request/cif.

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

## Acknowledgements

The authors are grateful for the funding support from the National Natural Science Foundation of China Grant (21788102), the Research Grants Council of Hong Kong (16304819, 16307020, C6014-20W, N_HKUST609/19 and 16305320), the Innovation and Technology Commission (ITC-CNERC14SC01), and the Natural Science Foundation of Guangdong Province (2019B121205002 and 2019B030301003). H.Z. thanks to the support from the Fundamental Research Funds for the Central Universities (2021QNA4032) and the Open Fund of Guangdong Provincial Key Laboratory of Luminescence from Molecular Aggregates, and the South China University of Technology (2019B030301003).

## Author contributions

J.Z., P.A., and S.Z. contribute equally to this work. J.Z., H.Z., and B.Z.T. conceived and designed the experiments. J.Z. and P.A. performed the synthesis. J.Z., P.A., and H.S. did the photophysical measurements and analyzed the data. J.Z. and L.H. conducted theoretical calculations. H.H.Y.S. and I.D.W. performed the single-crystal measurements. S.Z. did the fabrication and spectra measurements of devices. J.Z., J.S., J.W.Y.L., H.Z., and B.Z.T. took part in the discussion and gave important suggestions. J.Z. and H.Z. co-wrote the paper. All authors approved the final version of the manuscript.

## Competing interests

The authors declare no competing interests.
