## [Peer Review File · Nature Communications]

Secondary through-space interactions facilitated single-molecule white-light emission from clusteroluminogensReviewers' Comments:

Reviewer #1:

Remarks to the Author:

In this work, Tang et al. introduced a new strategy to construct single-molecule white-light emission from non-conjugated small molecules by combing the multiple through-space interactions (TSI) and phosphorescence. Through photophysical study, theoretical calculation and crystal structure analysis, they deciphered the multiple emissions from through-space conjugation, through-space charge transfer (TSCT), and room-temperature phosphorescence, respectively. The experimental and theoretical results are solid to support the nature of intramolecular TSCT instead of intermolecular behaviors. Besides, they also highlighted the excited-state molecular motion which should be an important contribution to facilitating TSCT effect. At last, white-light emitters with good criteria of CIE coordinate and CCK were fabricated.

Despite TSI is of crucial importance to the photoluminescence of non-conjugated luminophores, its manipulation remains challenging. Meanwhile, nonconjugated compounds with distinct white light emissions are rare. This work seems interesting, which provides a clear structure-property relationship of TSI and a general design strategy to manipulate emissions from single-component and non-conjugated molecules. Furthermore, the novel concept of secondary TSI is interesting, which might be an important photophysical mechanism for non-conjugated fluorophores and clusteroluminogens with isolated D-A structures. Based on these considerations, I recommend it for publication subject to some minor revisions.

1. The authors discussed the restriction of intramolecular motions of TPMI-X in the crystalline state and excited-state molecular motion to promote TSCT effect. Did the authors consider the two effects are contradictory? Upon freezing to 77 K, whether the TSCT effect can be impeded?
2. For Hirshfeld surface analysis, it is suggested to present all the decomposed fingerprint plots of intermolecular interactions, including other interactions mentioned in Figure 4g.
3. Although some references about TSCT were cited as Ref. 17, The authors may describe and compare the related research works when discussing TSCT effect.
4. In Figure 5f, the authors draw the schematic diagram to summarize the mechanisms for multiple emissions. Although the intramolecular TSCT process is clear according to experiments and hole-electron analysis, a complete TSC-based unit should be drawn to show its TSCT process, rather than a single N atom.
5. Why the long-wavelength fluorescence is slightly stronger than the short-wavelength fluorescence in these fabricated white-light emitters, compared with their PL spectra of crystals? Besides, is it possible to utilize them as emissive layers for OLED devices?
6. Calculation method or formula for RMSD analysis (Figure 5e) should be provided in Supporting Information.

Reviewer #2:

Remarks to the Author:

In the present manuscript, the authors reported the synthesis and characterization of new clusteroluminogens based on triphenylmethanamine and phenylmethanimine groups connected by a flexible ethyl chain. Three molecules were synthesized bearing different halogen atoms (I, Br and Cl) in the para position of the phenylmethanimine moiety. Such new molecules represent the goal of the new strategy proposed by the authors in order to prepare non-conjugated single-molecule with white-light emission features. The results here reported are consistent, novel and all the investigations suggest the role of the clusteroluminescence to promote the AIE response. The literature is up to date and contains consistent references that describe the main topic. I have only a few concerns concerning the experimental part:

- 1) at 350 nm the absorption of the three molecules is very low and results negligible at some extent, not only in solution but also in the solid film. How did the authors select the different excitation

wavelengths?

2) the authors did not provide photos of the films to determine the roughness of the surface. Is the surface homogeneous? Does the surface contribute to light scattering? What is the potential influence of light scattering on the overall emission characteristics?

3) the information concerning the realization of the silicone based composites and the LED construction are missing. Notably, the authors should evidence of the microcrystals were embedded in the silicone or epoxy matrix Did the embedding procedure affect the crystallinity degree of the molecules? Did the molecule dissolve in the silicone or epoxy matrix? Since the materials are characterized by a certain viscosity (that increases during curing for epoxy), how could the molecule be able to generate crystallites and enable clusteroluminescence?

Reviewer #3:

Remarks to the Author:

This manuscript by Tang and co-workers reported an interesting strategy to fabricate single-molecule white-light emission from a series of poorly conjugated compounds with isolated phenyl rings (clusteroluminogens, CLgens) in the crystalline state. Abnormal visible emission from nonconjugated structures (clusteroluminescence, CL) is a newly emerged phenomenon in the field of luminescent materials, which shows great theoretical significance. However, it seems that the underlying emission mechanism of CL is a thorny issue in this area. In this work, the proposed mechanism of secondary through-space interaction (TSI) for these CLgens is quite interesting, and different interactions (TBC, primary TSI and secondary TSI) are proven and elaborated through comprehensive molecular design, structural determination, photophysical characterization and theoretical calculation. Overall, this work seems significant which provides a new approach to manipulating through-space interactions and clusteroluminescence. Besides, the experiments are well performed, and the manuscript is well written. So, this manuscript is recommended to be published in Nature Communications after addressing the following minor points:

(1) In the introduction part, the authors mentioned that CLgens showed good biocompatibility compared with largely conjugated luminophores (Line 50). Detailed reasons and related references are suggested to add to the main text.

(2) The authors compared the emission and excitation spectra of both pure TPMA-X compounds and physical mixtures of Me-TPMA/Me-PMI-X, supporting their claim that the long-wavelength emission (or through-space charge transfer) was an intramolecular CT. From these results, I basically agree with the claim. But I am wondering whether the intermolecular CT process also contributed to the long-wavelength emission since the spectrum shown in Fig. 2h showed a small peak at about 450 nm.

(3) Please provide a discussion about the role of the amino group in triphenylmethanamine (TPMA) unit. Although the authors discussed the excited-state molecular motion between two separated D-A units as shown in Fig. 5b-c and 5f, I am curious about what will happen if no amino group is present in the TPMA unit.

(4) Could the authors explain why the relative PL intensity (I_L/I_S) in TPMA-I is higher than that of TPMA-Br and TPMA-Cl under the same excitation wavelength (Fig. 2g)? Is it related to the halogen atom?

(5) Regarding the application of white-light emitters, although it's just a simple coating of compounds on a UV lamp, it is acceptable as these non-conjugated compounds are hardly utilized in the emissive layer of OLED devices due to their poor charge transportability. A simple discussion about it should be added in this part to illustrate the advantages and drawbacks of such clusteroluminogens.

(6) The paragraph above Fig. 1 (Line 112), "All the spectra showed two peaks in the visible range". It is not accurate since there is a third peak of RTP discussed in the next part (although it is very weak compared with fluorescence).

(7) Regarding the theoretical calculations of energy levels and SOC constants, did the authors consider their differences in the isolated phase and crystalline phase, although the results were presented in Fig. S35?

(8) The terms "isolated phase" (Fig. S35) and "solution phase" (Fig. S37) are not completely clear. Are they the same? If yes, please use an accurate and unified term.

Response Letter to Reviewers

Response to the Comments of Reviewer 1:

In this work, Tang et al. introduced a new strategy to construct single-molecule white-light emission from non-conjugated small molecules by combing the multiple through-space interactions (TSI) and phosphorescence. Through photophysical study, theoretical calculation and crystal structure analysis, they deciphered the multiple emissions from through-space conjugation, through-space charge transfer (TSCT), and room-temperature phosphorescence, respectively. The experimental and theoretical results are solid to support the nature of intramolecular TSCT instead of intermolecular behaviors. Besides, they also highlighted the excited-state molecular motion which should be an important contribution to facilitating TSCT effect. At last, white-light emitters with good criteria of CIE coordinate and CCK were fabricated.

Despite TSI is of crucial importance to the photoluminescence of non-conjugated luminophores, its manipulation remains challenging. Meanwhile, nonconjugated compounds with distinct white light emissions are rare. This work seems interesting, which provides a clear structure-property relationship of TSI and a general design strategy to manipulate emissions from single-component and non-conjugated molecules. Furthermore, the novel concept of secondary TSI is interesting, which might be an important photophysical mechanism for non-conjugated fluorophores and clusteroluminogens with isolated D-A structures. Based on these considerations, I recommend it for publication subject to some minor revisions.

Response: We highly appreciate the reviewer for his/her positive assessments of our work. According to the valuable suggestions from the reviewer, we have carefully revised the manuscript.

1. The authors discussed the restriction of intramolecular motions of TPMI-X in the crystalline state and excited-state molecular motion to promote TSCT effect. Did the authors consider the two effects are contradictory? Upon freezing to 77 K, whether the TSCT effect can be impeded?

Response: Thanks for your comments. In fact, these two effects are not contradictory. For one thing, compared with violent intramolecular motions in the solution state, intramolecular motions are restricted to a certain extent (not wholly restricted) in the crystalline state. These motions are comparatively strong in the solution state, which induce significant geometry change and are responsible for the non-radiative decay. Hence, the restriction of intramolecular motions (RIM) in the crystalline state can partially block the non-radiative decay and promote clusteroluminescence (CL). On the other hand, excited-state molecular motion (ESMM) is comparatively weaker in crystalline than that in the solution state, which slightly shortens the distance between two N atoms. As shown in Fig. 5e, this motion is not entirely restricted in the crystalline state (RMSD = 0.326 Å). Therefore, the ESMM can facilitate the TSCT process rather than cause non-radiative decay. In summary, the RIM and ESMM effects coexist in the crystalline state, which synergistically promotes the TSCT and boosts the CL.

Upon freezing to 77K, the TSCT effect is not completely impeded since TPMI-X crystals show clearly blue-shifted TSCT emission (~470 nm) than that at room temperature (~500 nm) (Fig. S23 & Fig. 2a-c), suggesting partially limited molecular motions at low temperature. Besides, we also measured the PL spectra of TPMI-X in the solution state at 77 K (Fig. R1). Due to the much flexible conformation, the TSCT emission is comparatively weaker than the emission of the TPMA group with a wavelength of 370 nm. These results suggest that apart from ESMM, the TSCT effect is also affected by structural flexibility (RIM effect), which

further confirms the synergistic effect of RIM and ESMM on the CL.

Fig. R1 Photoluminescence spectra of **a** TPMI-Cl, **b** TPMI-Br, and **c** TPMI-I in THF solution at 77 K. Concentration = 10^{-4} M, $\lambda_{\text{ex}} = 310$ nm.

2. For Hirshfeld surface analysis, it is suggested to present all the decomposed fingerprint plots of intermolecular interactions, including other interactions mentioned in Figure 4g.

Response: Thanks for your good suggestion. We have analyzed all the intermolecular interactions and presented the corresponding decomposed fingerprint plots in Fig. S30 in the Supporting Information.

Fig. S30 Hirshfeld surfaces (mapped over d_{norm}) and decomposed fingerprint plots **a** TPMI-Cl, **b** TPMI-Br, and **c** TPMI-I. Full fingerprints appeared as grey shadows underneath decomposed plots, and selected intermolecular interactions were shown as a blue shadow. The proportions of different intermolecular interactions to the total interactions are indicated.

3. Although some references about TSCT were cited as Ref. 17, The authors may describe and compare the related research works when discussing TSCT effect.

Response: Thanks for your good suggestion. Through-space charge transfer (TSCT) is a spatial charge transfer process between two nonconjugated units, which is widely studied and proved as a new mechanism for luminescence from some organic molecules in the field of thermally activated delayed fluorescence and RTP (Ref. 44-49). Usually, the subunits for TSCT have π -conjugated structures and a face-to-face arrangement, facilitating the charge transfer process. However, the TSCT studied in this work is occurred in the subunit with a non-conjugated TSC (through-space conjugation) structure, which is different from all the reported TSCT systems.

To better compare the TSCT effect with the reported studies, we have added and revised the related description in the main text: “The effect of TSCT is widely studied and proved as a new mechanism for luminescence from conjugated organic luminogens, especially in the field of thermally activated delayed fluorescence and RTP. It is noteworthy that, different from the reported TSCT process caused by the spatially electronic interactions between the TBC-based donor and acceptor units, the TSCT in TPMI-Br is based on the non-conjugated donor with TSC.”

4. In Figure 5f, the authors draw the schematic diagram to summarize the mechanisms for multiple emissions. Although the intramolecular TSCT process is clear according to experiments and hole-electron analysis, a complete TSC-based unit should be drawn to show its TSCT process, rather than a single N atom.

Response: Thanks a lot for your comment. According to your suggestion, we have revised the diagram and presented the complete TSC-based unit in Fig. 5f to better show the behaviors of TSCT between two non-conjugated units.

5. Why the long-wavelength fluorescence is slightly stronger than the short-wavelength fluorescence in these fabricated white-light emitters, compared with their PL spectra of crystals? Besides, is it possible to utilize them as emissive layers for OLED devices?

Response: Thanks for your comments. Two reasons may account for the slightly stronger long-wavelength fluorescence: (1) we utilized a commercial UV chip with an emission maximum at 310 nm to fabricate these emitters, thus the short-wavelength excitation may be beneficial for emission from the TSCT process compared to the PL spectra excited using excitation of 350-390 nm; (2) the microcrystals of TPMI-X were mixed with silicone at 5 wt.% to fabricate these emitters. Thus, the matrix of silicone may slightly influence its luminescent spectra and relative intensity of the two peaks.

For these three compounds, it is difficult to solely utilize them as the emissive layer for

real OLED devices since their poorly conjugated structure and low ability of charge transport. In this work, we fabricated white-light emitters as an example to show their multiple emissions and a potential application of clusteroluminogens. For emissive layers in OLED devices, the active material features electroluminescence upon recombination of charged carriers injected at the electrodes, thus requiring efficient charge transport and exciton confinement based on the traditional conjugated structures (*Adv. Funct. Mater.* 2021, 31, 2100704; *Nat. Commun.* 2020, 11, 2732.). Hence, the addition of materials for enhancing charge transport is needed to utilize these compounds as emissive layers in OLED, which is considered in our future work.

To better illustrate the present drawbacks of such clusteroluminogens, we have added a discussion in this part that “It is worth noting that the poor ability of charge transport due to the non-conjugated structure hinders their application as an emissive layer in organic light-emitting diodes, although they displayed multiple emissions and good color index.”

6. Calculation method or formula for RMSD analysis (Figure 5e) should be provided in Supporting Information.

Response: Thanks a lot for your kind suggestion. We have added the calculation method for RMSD analysis in the method part of the manuscript:

“Root-mean-square deviation. Root-mean-square deviation (RMSD) was calculated to evaluate the strength of intramolecular motions between the optimized ground-state and excited-state geometries. The definition of RMSD is:

$$RMSD = \sqrt{\frac{1}{N} \sum_i^{natom} [(x_i - x'_i)^2 + (y_i - y'_i)^2 + (z_i - z'_i)^2]}$$

Where x , y , and z are the cartesian coordinates of each atom based on the optimized ground-state geometry (x_i , y_i , z_i) and excited-state geometry (x'_i , y'_i , z'_i), respectively.”

Response to the Comments of Reviewer 2:

In the present manuscript, the authors reported the synthesis and characterization of new clusteroluminogens based on triphenylmethanamine and phenylmethanimine groups connected by a flexible ethyl chain. Three molecules were synthesized bearing different halogen atoms (I, Br and Cl) in the para position of the phenylmethanimine moiety. Such new molecules represent the goal of the new strategy proposed by the authors in order to prepare non-conjugated single-molecule with white-light emission features. The results here reported are consistent, novel and all the investigations suggest the role of the clusteroluminescence to promote the AIE response. The literature is up to date and contains consistent references that describe the main topic. I have only a few concerns concerning the experimental part:

Response: Thanks a lot for the reviewer's recognition of the novelty and importance of our work. We have revised the manuscript according to your valuable suggestions.

1) at 350 nm the absorption of the three molecules is very low and results negligible at some extent, not only in solution but also in the solid film. How did the authors select the different excitation wavelengths?

Response: Thanks a lot for the comment. The excitation wavelengths are selected based on the excitation spectra of solid-state samples that showed intensity maxima at around 370 nm (Figs. 2h and S21). For clusteroluminogens with poorly conjugated structures, the absorption spectra only show the short-wavelength absorption from TBC-based units. And the effect of through-space conjugation and interactions (TSC and TSI) was only observed in the excited state, as evidenced by the electron overlaps at the LUMO (*J. Am. Chem. Soc.* 2017, 139, 45, 16264; *JACS Au* 2021, 1, 11, 1805; *J. Am. Chem. Soc.* 2021, 143, 25, 9565). Therefore, the corresponding excitation maxima could be determined from the excitation spectra.

2) the authors did not provide photos of the films to determine the roughness of the surface. Is the surface homogeneous? Does the surface contribute to light scattering? What is the potential influence of light scattering on the overall emission characteristics?

Response: Thanks a lot for your comment. In this work, we studied the photophysical properties of the three compounds in the crystalline state and utilized several model compounds to find their luminescent mechanism. Apart from the absorption measurement for thin films, all the emission characteristics (Figs. 2-3) and related discussions presented in the manuscript were based on the crystalline state instead of films. The photos of their crystals were also provided in Figs. 2 and S18, and the crystallinity degree of these bulk crystals was verified by the powder X-ray diffraction analysis (Fig. S22). For light scattering, it usually shows as Raman scattering in the emission spectra whose peak is narrow and excitation dependent. In our experiments with excitation wavelengths of 350-390 nm, we didn't observe such narrow and excitation-dependent peaks from light scattering. Therefore, it is suggested that these emission spectra show the intrinsic characteristics of these crystals, which are not affected by light scattering.

As for the application of white-light emitters, they are fabricated by mixing microcrystals of TPMI-X with silicone at 5 wt.% (Fig. R2). Therefore, the coating layer of these mixtures is thick, and the surface of the coating layer on UV chips is rough and not homogeneous. However, we can't observe any elastic scattering as narrow peaks from the emission spectra of these fabricated emitters (Figs. 6, S38, and S39). Therefore, it is suggested that their emission is also not influenced by light scattering of the surface.

Fig. 2R The photos of fabricated white-light emitters under room light.

3) the information concerning the realization of the silicone based composites and the LED construction are missing. Notably, the authors should evidence of the microcrystals were embedded in the silicone or epoxy matrix Did the embedding procedure affect the crystallinity degree of the molecules? Did the molecule dissolve in the silicone or epoxy matrix? Since the materials are characterized by a certain viscosity (that increases during curing for epoxy), how could the molecule able to generate crystallites and enable clusteroluminescence?

Response: Thanks for your careful review and comment. The details about the fabrication of these white-light emitters are provided in the method part of the manuscript (page 14).

“White-light emitters. The microcrystals of organic compounds (TPMI-X, X = Cl, Br, and I) were mixed with silicone at 5 wt.%, and the mixture was manually stirred with a stainless-steel bar for approximately five minutes until it was uniform. Then the mixture was coated on the surface of a commercial UV chip until the reflective cavity was filled. These chips were purchased from the Shenzhen Looking Long Technology Co., LTD. (Shenzhen, China) with a power of 1 W and an emission maximum of 310 nm. The performance of the fabricated emitters was recorded using an auto-temperature LED optoelectronic analyzer with a temperature controller (ATA-500, EVERFINE, China).”

According to the procedure described above, the bulk crystals are lightly ground into microcrystals and mixed with silicone. During the 12-hour curing process, these microcrystals don't dissolve in the silicone and can be clearly observed (Fig. R2). To verify their crystallinity in the silicone matrix, we fabricate them on a glass plate and do a powder X-ray diffraction (PXRD) analysis (Fig. R3). Apart from the board peak from the glass plate, three mixtures show clear diffraction peaks from specific crystal faces, indicating their high degree of crystallinity. And their intensity difference is from the degree of grinding before mixing with silicone. Therefore, our fabrication procedure and PXRD data support their state of crystal in the silicone mixtures, which further produce clusteroluminescence upon excitation of the UV chip.

Fig. R3 Powder X-ray diffraction patterns of silicone and mixtures of TPMI-Cl, TPMI-Br, and TPMI-I on a glass plate, respectively.

Response to the Comments of Reviewer 3:

This manuscript by Tang and co-workers reported an interesting strategy to fabricate single-molecule white-light emission from a series of poorly conjugated compounds with isolated phenyl rings (clusteroluminogens, CLgens) in the crystalline state. Abnormal visible emission from nonconjugated structures (clusteroluminescence, CL) is a newly emerged phenomenon in the field of luminescent materials, which shows great theoretical significance. However, it seems that the underlying emission mechanism of CL is a thorny issue in this area. In this work, the proposed mechanism of secondary through-space interaction (TSI) for these CLgens is quite interesting, and different interactions (TBC, primary TSI and secondary TSI) are proven and elaborated through comprehensive molecular design, structural determination, photophysical characterization and theoretical calculation. Overall, this work seems significant which provides a new approach to manipulating through-space interactions and clusteroluminescence. Besides, the experiments are well performed, and the manuscript is well written. So, this manuscript is recommended to be published in Nature Communications after addressing the following minor points:

Response: We appreciate the positive assessment from the reviewer. We have revised the manuscript according to your valuable suggestions.

(1) In the introduction part, the authors mentioned that CLgens showed good biocompatibility compared with largely conjugated luminophores (Line 50). Detailed reasons and related references are suggested to add to the main text.

Response: Thanks for your good suggestion. Compared with inorganic materials such as carbon nanotubes and perovskite, organic molecules show low toxicity and better degradability (*Nat. Nanotechnol.* 2020, 15, 3; *Chem. Soc. Rev.*, 2022, 51, 1983). Besides, compared with largely conjugated luminophores, small organic molecules with non-conjugated units exhibit a faster degradation rate with fewer reaction steps, resulting in better biocompatibility (*Chem. Soc. Rev.*, 2011, 40, 2673; *Chin. J. Polym. Sci.* 2015, 33, 680). Therefore, in the introduction part, we have revised the description as “Compared with traditional luminophores with conjugated aromatic rings, CLgens possess better flexibility and processability. Besides, they show good biocompatibility compared with inorganic materials due to the low toxicity and better degradability, thus serving as promising luminescent materials for biological applications. (Ref. 17-18)”.

(2) The authors compared the emission and excitation spectra of both pure TPMI-X compounds and physical mixtures of Me-TPMA/Me-PMI-X, supporting their claim that the long-wavelength emission (or through-space charge transfer) was an intramolecular CT. From these results, I basically agree with the claim. But I am wondering whether the intermolecular CT process also contributed to the long-wavelength emission since the spectrum shown in Fig. 2h showed a small peak at about 450 nm.

Response: Thanks a lot for your careful review and comment. By comparing the

emission and excitation spectra of pure TPMI-X compounds and mixtures of Me-TPMA/Me-PMI-X, we can find some differences between intramolecular CT and intermolecular CT. For the intramolecular CT, the emission maximum locates at around 500 nm, and the excitation maximum locates at 370 nm, respectively (Figs. 2 and S21). Besides, for the intermolecular CT process, the emission maximum is around 530 nm, and the excitation maximum is about 460 nm, respectively (Figs. 4 and S26). The above results suggested that higher excitation energy is needed to promote the intramolecular CT (with a shorter emission wavelength) from pure TPMI-X crystals. As mentioned by the review, a small peak at about 450 nm is observed in the excitation spectra of TPMI-Br with the corresponding emission at 500nm, which is close to the excitation maximum for intermolecular CT. However, the intensity of this peak is much lower than that at 370 nm. Therefore, it is reasonable that the long-wavelength emission from pure TPMI-X crystals is mainly contributed by the intramolecular CT process, while the intermolecular CT process plays a minor role in the long-wavelength emission. Accordingly, we have revised the related claim in the manuscript: “Therefore, it was believed that the long-wavelength emission from TPMI-X was mainly contributed by the intramolecular behaviors of two separated units, while the intermolecular CT process may play a minor role.”

(3) Please provide a discussion about the role of the amino group in triphenylmethanamine (TPMA) unit. Although the authors discussed the excited-state molecular motion between two separated D-A units as shown in Fig. 5b-c and 5f, I am curious about what will happen if no amino group is present in the TPMA unit.

Response: Thanks for your careful review and great suggestion. The amino group in the TPMA unit should play an essential role in promoting the TSCT process and producing the long-wavelength fluorescence. For one thing, the amino group is an electron-rich unit. Therefore, it increases the electron-donating ability of the TPMA part. In addition, the large distribution area of the lone pair enhances the electron overlap between the two separated D-A units, which facilitates the through-space interaction. Accordingly, the long-wavelength emission from TSCT would be weak or disappear if no amino group is present in the TPMA unit. Therefore, we have added a sentence to discuss its role in the manuscript: “Besides, the amino group of TPMA unit played as a “bridge” to facilitate the secondary TSI and increase the electron-donating ability of TPMA unit.”

(4) Could the authors explain why the relative PL intensity (IL/IS) in TPMI-I is higher than that of TPMI-Br and TPMI-Cl under the same excitation wavelength (Fig. 2g)? Is it related to the halogen atom?

Response: Thanks for your comment. From the photophysical measurements and crystal structure analysis, we speculate that this phenomenon is related to halogen atoms. The crystal structures of the three compounds possess almost the same intermolecular interactions (Figs. 4g and S27-29), and each halogen atom forms two halogen...H bonds with the TPMA unit of another molecule. It is acknowledged that halogen atoms could

facilitate non-radiative decay through the heavy-atom effect and finally decrease the efficiency or quench the emission (*Nat. Rev. Mater.* 2020, 5, 869; *J. Phys. Chem. Lett.* 2019, 10, 595; *J. Am. Chem. Soc.* 2019, 141, 1010). And the degree of heavy-atom effect gradually increases from Cl, Br, to I. Therefore, it is reasonable that these halogen atoms in TPMI-X may quench the emission from TPMA units. As a result, the highest quantum yields of the three compounds decline from TPMI-Cl (34.8%), TPMI-Br (28.3%), to TPMI-I (14.6). Besides, the intensity of the short-wavelength emission from TPMA units gradually decreases, and the relative intensity (I_L/I_S) increases from TPMI-Cl, TPMI-Br, to TPMI-I under the same excitation wavelength (Fig. 2g).

(5) Regarding the application of white-light emitters, although it's just a simple coating of compounds on a UV lamp, it is acceptable as these non-conjugated compounds are hardly utilized in the emissive layer of OLED devices due to their poor charge transportability. A simple discussion about it should be added in this part to illustrate the advantages and drawbacks of such clusteroluminogens.

Response: Thanks for your comment and suggestions. To show their multiple emissions and a potential application of clusteroluminogens in this work, we fabricated white-light emitters by coating our compounds with silicone on a commercial UV chip. It is a problem that such compounds are difficult to be used as the emissive layer for OLED devices since their poorly conjugated structure and low ability of charge transport. For emissive layers in OLDE devices, the active material features electroluminescence upon recombination of charged carriers injected at the electrodes, thus requiring efficient charge transport and exciton confinement based on conjugated structures (*Adv. Funct. Mater.* 2021, 31, 2100704; *Nat. Commun.* 2020, 11, 2732.). Hence, adding other materials to increase charge transport capability is needed to utilize these compounds as the emissive layer in OLED devices. Therefore, to better illustrate the drawbacks of such clusteroluminogens, we have added a discussion in this part “It is worth noting that the poor ability of charge transport due to the non-conjugated structure hinders their application as an emissive layer in organic light-emitting diodes although they displayed multiple emissions and good color index.”

(6) The paragraph above Fig. 1 (Line 112), “All the spectra showed two peaks in the visible range”. It is not accurate since there is a third peak of RTP discussed in the next part (although it is very weak compared with fluorescence).

Response: Thanks for your good concern. These compounds indeed show photoluminescence with three components, including two fluorescence peaks and a weak phosphorescence (Fig. 3). To avoid misleading, we have revised the sentence as “All the spectra showed two distinct emission peaks in the visible range, where their relative intensity was excitation dependent.”

(7) Regarding the theoretical calculations of energy levels and SOC constants, did the authors consider their differences in the isolated phase and crystalline phase, although the results were presented in Fig. S35?

Response: Thanks for your comment. In this work, we calculated the energy levels and SOC constants of the three compounds in both solution and crystalline phases (Fig. S35). The energy levels of the two states are different that the S_1 state is close to T_2 state in the solution phase while S_1 state is close to T_1 state in the crystalline phase. This is because the crystal packing brings a different molecular conformation and restricts the intramolecular motions, which decreases the energy levels of both S_1 and T_1 states and changes their relative position. On the other hand, the SOC constants are similar in the two different phases. As a result, the channel for efficient intersystem crossing changes from S_1 to T_2 in the solution state to S_1 to T_1 in the crystalline state. Apart from the RIM mechanism, the change in energy levels also promotes the room-temperature phosphorescence in the crystalline state.

(8) The terms “isolated phase” (Fig. S35) and “solution phase” (Fig. S37) are not completely clear. Are they the same? If yes, please use an accurate and unified term.

Response: Thanks for your careful review. Both the “isolated phase” and “solution phase” represent that the molecules are in the single-molecule state, which is different from the aggregate or crystalline state. To calculate compounds in the single-molecule state, they were optimized using a polarizable continuum model and self-consistent reaction field with tetrahydrofuran as solvent. Hence, these two terms are the same in the manuscript.

To provide a clear description, we have revised the term in Fig. S35 and utilized the “solution phase” in the whole manuscript and Supporting Information.

Fig. S35 Energy diagram and SOC values of the targeted compounds in the isolated and crystalline phases, calculated by TD-DFT method, B3LYP-D3/Def2-SVP.

Reviewers' Comments:

Reviewer #1:

Remarks to the Author:

My previous concerns have been fully addressed by the authors. This version is acceptable. Meanwhile, several minor points require further correction: 1) "CDCl₃", in which "3" should be subscripted; 2) For the fabrication of the LED device, whether "epoxy resin glue" or "silicone" was used? This should be double checked and corrected.

Reviewer #2:

Remarks to the Author:

The authors provided a consistent reply letter and most of the raised comments were appropriately considered. Nevertheless, I am not able to determine the presence of microcrystals from Fig. R2. Could the authors provide a microscopic image or a picture at higher resolution and magnification? Then, I suggest also the authors to add in the revised text the discussion provided about the presence of microcrystals in the silicone matrix. It may be useful for the readers. After these (very) minor revisions the paper can be accepted.

Reviewer #3:

Remarks to the Author:

The authors have well addressed my concerns. I think the quality of the manuscript is significantly improved. Thus, I recommend the acceptance of the manuscript by the journal as is.

Response Letter to Reviewers

Response to the Comments of Reviewer #1:

My previous concerns have been fully addressed by the authors. This version is acceptable.

Response: We highly appreciate the positive acknowledgment of our revised manuscript. Thanks for your precious time and valuable suggestions during the review process.

Meanwhile, several minor points require further correction: 1) "CDCl₃", in which "3" should be subscripted; 2) For the fabrication of the LED device, whether "epoxy resin glue" or "silicone" was used? This should be double checked and corrected.

Response: Thanks for your comments. We have revised it using the subscript for the solvent (CDCl₃) used for NMR measurement.

For the fabrication of white-light emitters, epoxy resin glue was used for all devices. We have revised all the related descriptions using "epoxy resin glue" in the manuscript and Supplementary Information.

Response to the Comments of Reviewer #2:

The authors provided a consistent reply letter and most of the raised comments were appropriately considered.

Response: We highly appreciate the reviewer for the positive assessments of our work and the revised manuscript.

Nevertheless, I am not able to determine the presence of microcrystals from Fig. R2. Could the authors provide a microscopic image or a picture at higher resolution and magnification? Then, I suggest also the authors to add in the revised text the discussion provided about the presence of microcrystals in the silicone matrix. It may be useful for the readers.

Response: Thanks a lot for your concerns. According to your suggestion, we have added a microscopic image of these fabricated white-light emitters to show the presence of microcrystals as Fig. S40 in the Supplementary Information. Accordingly, we have added a discussion about the state of these compounds in the epoxy resin glue that "These microcrystals didn't dissolve during fabrication and were embedded into the glue (Fig. S40)."

Fig. S40 Photos of the fabricated white-light emitters under the optical microscope. Microcrystals of these compounds are embedded into the epoxy resin glue.

Response to the Comments of Reviewer #3:

The authors have well addressed my concerns. I think the quality of the manuscript is significantly improved. Thus, I recommend the acceptance of the manuscript by the journal as is.

Response: Thanks a lot for the reviewer's precious time, recognition, and support for the publication of our work.